# HYBRID CLASSIFICATION-REGRESSION ADAPTIVE LOSS FOR DENSE OBJECT DETECTION

## ABSTRACT

In object detectors, enhancing model performance hinges on the ability to simultaneously consider inconsistencies across tasks and focus on difficult-to-train samples. Achieving this necessitates incorporating information from both the classification and regression tasks. However, prior work tends to either emphasize difficult-to-train samples within their respective tasks or simply compute classification scores with IoU, often leading to suboptimal model performance. In this paper, we propose a Hybrid Classification-Regression Adaptive Loss, termed as HCRAL. Specifically, we introduce the Residual of Classification and IoU (RCI) module for cross-task supervision, addressing task inconsistencies, and the Conditioning Factor (CF) to focus on difficult-to-train samples within each task. Furthermore, we introduce a new strategy named Expanded Adaptive Training Sample Selection (EATSS) to provide additional samples that exhibit classification and regression inconsistencies. To validate the effectiveness of the proposed method, we conduct extensive experiments on COCO test-dev. Experimental evaluations demonstrate the superiority of our approachs. Additionally, we designed experiments by separately combining the classification and regression loss with regular loss functions in popular one-stage models, demonstrating improved performance.

## 1 INTRODUCTION

Over recent years, object detection has garnered significant attention and has been widely employed in domains like pedestrian detection (Dana et al. (2021); Zhou et al. (2023); Liu et al. (2018)) and face recognition (Fan & Jiang (2021)). It encompasses two primary tasks: classification and regression. It aims to predict classification scores and bounding-box coordinates based on input images with a large amount of background information, which lead to an imbalance between positive and negative samples. This imbalance makes it arduous for models, particularly one-stage detectors (Kim & Lee (2020); Zhu et al. (2019a); Zhang et al. (2019)), to focus training on relevant samples. This imbalance between the background and positive samples has propelled researchers to explore mechanisms that could enable models to concentrate more intently on the difficult-to-train samples. For instance, methods like Focal Loss (Lin et al. (2017)) and GHM Loss (Li et al. (2019)) were designed within classification tasks. Similarly, in regression task, methods such as Focal EIoU (Zhang et al. (2022)) and Alpha IoU (He et al. (2021)) emphasize difficult samples by modulating the gradient.

However, in recent years, researcher (Wu et al. (2020)) has found that redundant boxes processed by non-maximum suppression (NMS) may lead to the exclusion of certain boxes with high localization ability but low classification scores, thus reducing the model performance. This reminds us that the loss function design needs to consider both the consistency of classification scores and IoU. To address this issue, the Generalized Focal Loss (GFL) (Li et al. (2020)) function incorporates the IoU as a classification label, while Varfocal loss (Zhang et al. (2021)) proposes a cross-entropy function that incorporates localization information. However, both methods fail to effectively focus on truly difficult-to-train samples when dealing with samples with similar IoU. In addition, the loss functions of the IoU series (Yu et al. (2016); Zheng et al. (2020)) mostly ignore the consistency of classification and regression, and more often than not focus only on difficult-to-localize samples.

To address the above challenges, we propose the HCRAL (Hybrid classification-regression adaptive loss), which consists of a module that characterizes the consistency of classification and regression

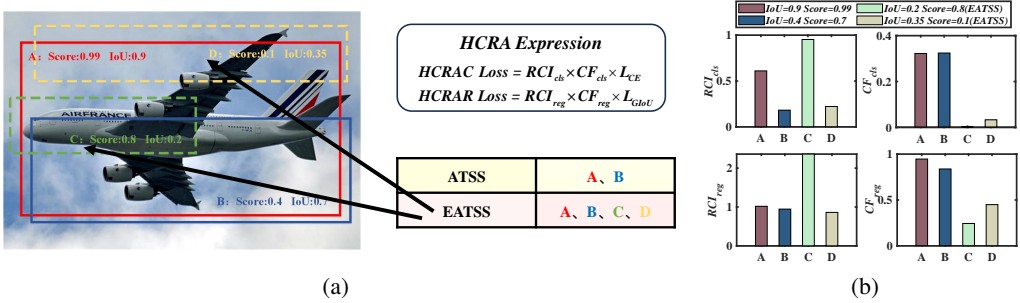

Figure 1: The diagram provides an overview of our HCRA loss composition and the positive-negative sample selection strategy. Both the classification and regression loss functions incorporate adaptive modules, such as the RCI (residual of cls and IoU) module and the CF (conditioning factor) module. In the context of ambiguous anchors A, B, C, and D, the values associated with the RCI module and CF module are visually represented in the bar chart. Notably, anchors C and D represent samples newly introduced by EATSS (Expanded Anchor Target Sampling Strategy) in comparison to ATSS. Please see more details about EATSS in Algorithm 1.

and is applied to both classification and regression tasks, namely residual of cls and IoU (RCI), and a conditioning factor (CF) that can be focused according to the difficult-to-train samples of the respective tasks, as shown in Figure 1(a). First, we use GHM loss (Li et al. (2019)) and GIoU (Rezatofighi et al. (2019)) loss as the basis functions and design RCI module, which provides mutual information for the classification and regression loss functions. In addition, we adjust attention for positive and negative samples in classification and difficult-to-train samples in regression. The distribution of RCI and CF in classification and regression for different samples is shown in Figure 1(b). Also, we propose a new positive and negative sample allocation strategy called Expand Adaptive Training Sample Selection (EATSS) to provide more samples with just high IoU or just high classification scores (C, D samples in Figure 1(a)) to optimize the loss function.

To better validate the effects of our HCRA loss, we incorporate it into a popular one-stage model, as illustrated in Figure 2. Additionally, we introduce HCRA loss and EATSS based FCOS+ATSS structure when compared with other loss methods. To further explore the performance of our methods, the star convolution and bounding box refinement components are applied as auxiliary modules. In this adaptation, we retain the centerness branch while altering the target to IoU scores instead of the original design.

Our main contributions can be summarized as follows:

- The HCRAL is proposed, which is a novel loss that focuses across tasks. It establishes an RCI module for models to supervise each other in the classification and regression tasks while CF modules focus on difficult-to-train samples within each task.
- To accommodate the proposed HCRA loss function, we introduce a new ATSS-based EATSS strategy to provide more optimizable positive samples to the RCI module.
- To demonstrate the superiority and generality of our loss function, our proposed HCRA loss is combined with different loss functions in popular one-stage models. We also show higher accuracy of our proposed methods based on FCOS+ATSS structure when compared to existing state-of-the-art loss functions on COCO test-dev.

## 2   RELATED WORK

**One-stage Object Detectors:** Unlike two-stage detectors (Ren et al. (2015); Cai & Vasconcelos (2018); He et al. (2017); Dai et al. (2016)), one-stage detectors directly predict classification probabilities and position coordinate offset, rather than generating manageable number of region proposals called region of interest (ROI) , which results in a fast detection speed. For one-stage detection models, it is generally divided into anchor-based and anchor-free. Anchor-based aims to generate classification and regression through anchor. Those models are classics such as Retinanet (Lin et al.

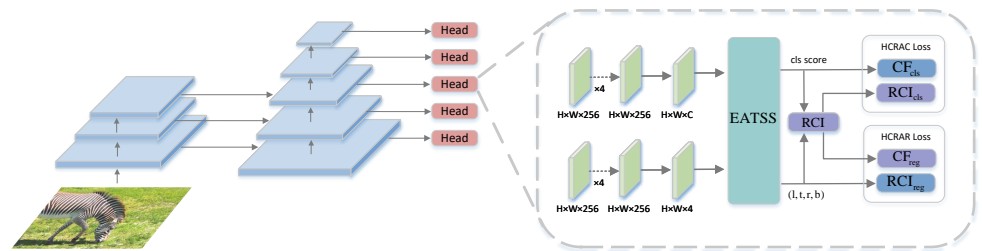

Figure 2: A diagram of structure of HCRAL. HCRAL includes CF and RCI. RCI, derived from the regression and classification predictions, aims to emphasize consistency. Meanwhile, CF is designed to focus on difficult-to-train samples. To futher exploring the performance of HCRAL, EATSS strategy is adopted.

(2017)) and SSD (Liu et al. (2016)). For anchor-free models, there are two ways to predict the position of objects, which offer flexibility and convenience: anchor-point prediction and key-point prediction. The key-point based model (Law & Deng (2018); Duan et al. (2019); Zhou et al. (2019)) predicts the target box and classifies it by predicting the corner points. Another anchor-point model similar to key-point generates the prediction area of the target more dynamically, and predicts the distance from the four boundaries of the target box through the information of the anchor-point itself, including FCOS (Tian et al. (2019)) and ATSS (Zhang et al. (2020)). In recent years, anchor free methods has also been used in many popular frameworks, such as the yolo series (Ge et al. (2021); Redmon et al. (2016); Redmon & Farhadi (2017)).

**Cost Functions of Object Detectors:** In the development of the object detection, the imbalance between positive and negative samples has always been a difficult problem to solve. For classification loss, Focal loss (Lin et al. (2017)) and GHM (Li et al. (2019)) loss are applied in one-stage model. To combine with IoU information, AP series (Xu et al. (2022)) loss aim to enhance the performance metrics, which still hard to optimize. While Varifocal loss (Zhang et al. (2021)) and GFocal (Li et al. (2020)) take IoU as classification lalels without considering difficult-to-train samples. For regression loss, existing IoU series loss function fall into two main methods, one of them (Rezatofighi et al. (2019); Zheng et al. (2020)) is to increase the penalty by increasing the centroid, width and height of the error. The other methods (Tong et al. (2023); Zhang et al. (2022); He et al. (2021)) is mainly to adjust the weights of high-quality examples and low-quality examples to increase the focus on difficult-to-train samples. However, above loss functions fail to consider both the consistency of IoU and score and difficult-to-train samples.

## 3 METHOD

Through above analysis, we introduce the RCI module for mutual supervision in classification and regression, along with the CF module for focusing on difficult-to-train samples in Section 3.1. We will present the new positive and negative sample selection strategy called EATSS in Section 3.2.

### 3.1 LOSS FUNCTION DESIGN

**Consistency of Classification and Regression.** As depicted in Figure 3(a), the points near the red line adhere to classification-regression consistency, but the majority of points do not. In the existing loss functions, there is limited consideration for simultaneously incorporating consistency into both classification and regression tasks. To tackle this, we introduce RCI module, enabling optimization of classification and regression based on the degree of inconsistency exhibited by each anchor point:

$$RCI = s - iou + \alpha \tag{1}$$

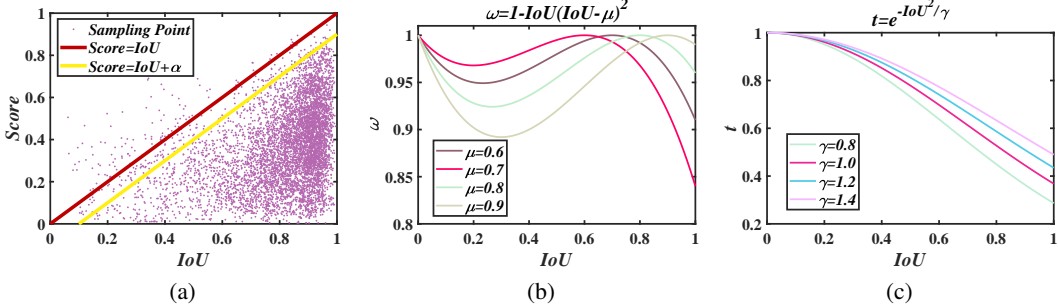

Figure 3: Illustrations of various curves of different parameter. (a) The distribution of IoU and classification scores. (b) The curves of $\omega$ with different $\mu$ when $p^* = 0$. (c) The curves of $t$ with different $\gamma$.

where $s$ is the predicted cls score and $iou$ is are the locations of the predicted bounding box and the ground truth in pos samples. $\alpha$ is a parameter that regulates the balance between classification and IoU (yellow lines in Figure 3(a)).

### 3.1.1 CLASSIFICATION LOSS FUNCTION

We analyze the existing classification functions based on experience. The detector can pay attention to difficult-to-classify samples, GHM loss shows good adaptive ability, can be written as:

$$
\begin{aligned}
L_{GHM-C} &= \frac{1}{N} \sum_{i=1}^{N} \frac{N}{GD(g_i)} L_{CE}(p_i, p_i^*) \\
&= \frac{1}{N} \sum_{i=1}^{N} \beta_i L_{CE}(p_i, p_i^*)
\end{aligned}
\tag{2}
$$

where $p$ is the predicted probability and $p^*$ is the ground-truth label. $N$ is the number of total samples. The gradient can be partitioned into $M$ subintervals, each with a length of $l_\varepsilon(g)$. $g_k$ is the center of the $k$-th subinterval. $\delta_\varepsilon(g_k, g)$ is a function that determines whether the sample gradient $g$ is satisfied in the $k$-th subinterval. $GD(g) = \frac{1}{l_\varepsilon(g_k)} \sum_{k=1}^{N} \delta_\varepsilon(g_k, g)$ is described as calculating the number of samples in each gradient subinterval. Where $\beta_i = \frac{N}{GD(g_i)}$ is the weight of the $i$-th sample. However, the classification loss function needs to have the following points so that all the training samples are effectively treated:

- For positive samples, the weights assigned to samples with high IoU should be greater than those samples with low IoU to prevent large penalties on low IoU samples, when in same subinterval.

- Further, samples with classification scores significantly lower than their corresponding IoU should receive higher attention. However, as the classification scores approach the IoU, the weights should decrease, potentially reaching zero. Beacause the model doesn't need to allocate excessive training focus on classification.

- For negative samples, the weighting logic differs. In cases samples with low IoU, indicating a greater potential for a false positive, which remain relatively high. As the IoU increases, these weights should gradually diminish, reflecting the decreasing risk of false positives.

To specify the above first and second conditions, we define adaptive matrix, denoted $\omega$, which can be expressed as:

$$
\omega =
\begin{cases}
IoU & if\ p^* = 1 \\
(1 - IoU(IoU - \mu)^2) & if\ p^* = 0
\end{cases}
\tag{3}
$$

$\mu$ is a sample weight decay factor that controls for a high IoU when $p^* = 0$, as can be seen in Figure 3(b). To satisfy the second point, we keep the gradient of GHMC and combine $\omega$ with it, namely conditioning factor (CF), which is defined as follows:

$$CF_{cls}(i) = \omega_i * \beta_i \tag{4}$$

To meet the third condition, we introduce the RCI module to achieve consistency between classification and IoU:

$$RCI_{cls} = \begin{cases} \frac{e^{\theta RIC}}{(e^{\theta RIC}+1)} & \text{if } p^* = 1 \text{ and } p > \text{IoU} \\ 0 & \text{if } p^* = 1 \text{ and } p \leq \text{IoU} \\ 1 & \text{if } p^* = 0 \end{cases} \tag{5}$$

$\theta$ is an adjustment factor for score and iou. When the score gets higher and closer to the iou, the consensus matrix will become smaller, so as to reduce attention of the model to the sample. When $\theta$ become larger, the more obvious the inhibitory effect. It can be observed that HCRAC comprises $RCI_{cls}$ and $CF_{cls}$ in Figure 2, which can be expressed as follows:

$$L_{HCRA-C} = CF_{cls}(i) * RCI_{cls}(i) * L_{CE}(p_i, p_i^*) \tag{6}$$

### 3.1.2 REGRESSION LOSS FUNCTION

We summarize the desired regression loss functions:

- The model needs to increase the gradient for some ordinary-quality anchor boxes.
- There are many low-quality samples with abnormal aspect ratios in the training data, which are difficult to produce high overlap with the groud truth in training and reduce the training quality.
- Low-score, high-IoU samples have limited learning potential, whereas high-score, low-IoU samples show the opposite. Therefore the model needs to increase the punishment for some low-iou anchor boxes accompanied by high scores, and reduce the gradient propagation for anchor boxes with low scores and high IoU.

Following the classification loss function, we separate all these conditions above from the computational graph to avoid the backpropagation will affect each other. The properties 1 and 2 are based on IoU to adjust the weight of the sample, we can construct conditioning factor as follows:

$$CF_{reg} = t \cdot e^{\mathcal{R}_{\text{DIoU}}} \cdot \text{IoU} \tag{7}$$

Here, $CF_{reg}$ represents the attention weight assigned to anchors that are both ordinary-quality and have a high IoU, and $\gamma$ is parameters in $t = e^{-\frac{\text{IoU}^2}{\gamma}}$ that govern the shape of the weight function for suppressing of high-iou anchor. Figure 3(c) shows the graph of $t$ about for different control parameters. $\mathcal{R}_{\text{DIoU}} = \frac{\rho(b, b^{gt})}{c^2}$ represents a certain degree of offset from the center point, and it can improve the model's attention to the unaligned anchors at the center point. $c$ is the diagonal length of the smallest enclosing box. $\rho$ represents calculating the distance between the center $b$ of the anchor box and the center $b^{gt}$ of the ground truth.

In conjunction with the third point above, we introduce the RCI module in order to focus on the samples with high IoU but low scores. As Figure 3(a) illustrates the distribution of scores and IoU, the coordinates are partitioned into two regions based on $y(Score) = x(IoU) + \alpha$. In region 1, where samples exhibit higher scores compared to IoU values, the model should prioritize its attention accordingly. Conversely, in region 2, the model's focus should decrease, as samples in this region have higher IoU values relative to their scores. We construct a RCI-based coefficient as:

$$RCI_{reg} = \begin{cases} \dfrac{(s+\alpha)^2 + IoU^2(tb, cb) + ep}{2*(s+\alpha)*IoU(tb, cb) + ep}, \text{if } RCI \geq 0 \\ \dfrac{2*(s+\alpha)*IoU(tb, cb) + ep}{(s+\alpha)^2 + IoU^2(tb, cb) + ep}, \text{if } RCI < 0 \end{cases} \quad (8)$$

Here, $(s+\alpha)^2 + IoU^2 >= 2*(s+\alpha)*IoU$, when $s, IoU \in [0, 1]$. Thus, when samples in region 1, $RCI >= 0$, which means $RCI_{reg} >= 1$. While samples in region 2 that satisfying $RCI < 0$, $RCI_{reg} \in [0, 1]$. $ep$ is an adjustment parameter that prevents $RCI_{reg}$ from being too large. We normalize $RCI_{reg}$ by running mean. The exponential moving average (EMA) method is applied to calculate weights in our work, which can be expressed as follows:

$$r^{(t)} = (1 - m)r^{(t-1)} + mRCI_{reg}^{(t)} \quad (9)$$

From Figure 2, we introduce r and $CF_{reg}$, utilizing GIoU as the foundational function. Consequently, our proposed HCRAR can be formulated as follows:

$$L_{HCRA-R} = \sum_{i=1}^{N} r \times CF_{reg} \times L_{GIoU}$$
$$L_{GIoU} = 1 - IoU + \frac{|C - (A \cup B)|}{|C|} \quad (10)$$

Where $IoU = \frac{|A \cap B|}{|A \cup B|}$. $A$, $B$ are two arbitrary boxes. $C$ is the smallest convex box enclosing both A and B.

### 3.2 EXPAND ADAPTIVE TRAINING SAMPLE SELECTION

---
**Algorithm 1** Expand Adaptive Training Sample Selection (EATSS)

---
**Require:** G is a set of ground-truth boxes on the image, L is the number of feature pyramid levels, $A_i$ is a set of anchor boxes from the $i$th pyramid levels, $A$ is a set of all anchor boxes, $k$ is a hyperparameter.
**Ensure:** P is a set of positive samples, N is a set of negative samples.
1: **for** each ground-truth $g \in G$ **do**
2:     Determine positive candidate anchors $C_g$ for g using ATSS.
3:     Filter candidates in $C_g$ based on IoU with $g$ and add to positive set $P$.
4:     Get the set $E$ satisfying the maximum distance $Dis_f$ of $P$.
5:     **for** each candidate $e \in E_g$ **do**
6:         Compute distance and IoU of $e$.
7:     **end for**
8:     Add selected l candidates from $E$ to $P$ based on distance and IoU.
9: **end for**
10: Determine negative set $N$ as anchors not in $P$.
11: **return** $P, N$;

---

While the adaptation of ATSS Zhang et al. (2020) algorithm facilitates the selection of k samples near the center of ground truth of each pyramid feature and dynamically identifies positive samples adhering to mean and variance of IoU, which may omit certain samples characterized by high scores and high IoU that hold promise.

Specifically, we need to increase effective positive samples, such as some high IoU and high score anchor, to optimize RCI for every group truth. As Algorithm 1 described, after the ATSS algorithm screening obtains positive and negative samples for each ground truth, find the maximum distance between centers of anchors and ground truth. $Dis_f$ that can meet sum of mean and variance of IoU, so as to find the biggest boundary to find a positive sample. In order to explore high IoU and high score anchor, we design a ranking function, consist of the distance between the center of the box to the center of the ground truth and IoU, to get the most positive sample screening l-point with the highest ranking score to provide more samples that can be optimized by RCI.

# 4 EXPERIMENTS

**Datasets.** We assess HCRAL and EATSS on the large-scale object detection dataset COCO (Lin et al. (2014)). Following common practice, we train detectors on the train2017 split, report ablation results on the val2017 split, and compare with other detectors on the test-dev split by uploading the results to the evaluation server. We adopt the standard COCO-style Average Precision (AP) as the evaluation metric.

**Experimental Setup.** To validate the efficacy of our loss function in two different one-stage detection methods, anchor-free and anchor-based, we choose ATSS and RetinaNet as detectors. Note that all loss weights of regression tasks are set as 2.5 for RetinaNet. Further, Due to the expandability of FCOS+ATSS structure, we selected it to combine our proposed loss function and EATSS. The initial learning rate is 0.01, and we implement the linear warm-up policy at the beginning of training, with a warm-up ratio (Goyal et al. (2017)) set to 0.1, except for ATSS, where the warm-up ratio is set to 0.001. Except for validating regression performance with 4 GPUs on RetinaNet, we use 8 V100 GPUs for training with a total batch size of 16 (2 images per GPU) in both ablation studies and performance comparisons. In particular, we conducted ablation studies on RetinaNet with backbone as ResNet-50 (He et al. (2016)) on COCO val2017 and trained FCOS+ATSS with different backbones on COCO test-dev. If the backbone utilizes DCN, it is also incorporated into the final layers preceding the star deformable convolution. When introducing auxiliary modules, we use the star convolution and bounding box refinement components into FCOS+ATSS. For fair comparison with state-of-the-art methods on COCO test-dev, 2x (24 epochs) training scheme and multi-scale training (MSTrain) are adopted.

## 4.1 ABLATION STUDIES

**Classification hyperparameters.** The tuning of positive samples relies on two hyperparameters: RCI and CF. The hyperparameter of M control the number of gradient subintervals. $\theta$ is used to regulate the weights of positive samples when score is less than IoU. $\gamma$ is used to adjust the relative weights of the negative samples. In Table 1 we show parameters $\theta$ from 4 to 6 and parameters $M$ from 20 to 30. it can be summarized that the model preforms best when $\theta$ is set to 5 and $M$ is set to 20. In the third line, 0.2 AP is drop without RCI module. This illustrates the effectiveness of RCI in classification tasks. From the results in Table 2, we choose the parameter $\gamma$ of 0.7 as the optimal parameter.

**Regression hyperparameters.** For regression analysis, there are three key hyperparameters. RCI modules, consists of two hyperparameters, $\alpha$ and ep. Parameter $\alpha$ represents the gap between classification and regression. When this gap increases, the consistency problem will become more significant because the number of samples located in region 1 will increase accordingly. On the other hand, the hyperparameter ep is used to control for the effects of classification scores and IoU inconsistency. Specifically in region 1, a decrease in the parameter ep leads to an increase of $RCI_{reg}$ to enable model focus on the inconsistency between classification and regression, while an increase in ep reduces the attention of inconsistency. Table 3 shows that the two parameters of RCI, $\alpha$ and ep, work best when taken 0.1, 0.001. It can observed that without RCI module, 0.3 AP is drop. This illustrates the effectiveness of RCI in regression tasks. It can be observed in Table 4 that $\gamma$ is set as 1.2 to get good performance. Due to the different samples selection in RetinaNet and ATSS, ATSS tends to include more low-IoU samples. To address this, we removed the $\gamma$ when using the ATSS and set set the weight to 1.5, ensuring that high-quality samples are not suppressed during training.

**EATSS.** Based on FCOS+ATSS with auxiliary modules and using ResNet-50 as the backbone, we combine HCRAL with EATSS to explore the effects of the l-additional anchor point. The parameter l controls the number of anchor points are added to each ground truth. Table 6 indicates that varying this parameter results in AP values ranging from 41.7 AP to 42.1 AP, demonstrating the sensitivity of performance of our HCRA loss to parameter l.

## 4.2 EVALUATION OF INDIVIDUAL METHOD CONTRIBUTIONS

To verify the effectiveness of individual methods, we add them in turn and summarize them in Table 5. Firstly, it can be seen that replacing EATSS with ATSS, only 0.1 AP increase for FL+GIoU. However, when the additional sample quantity (l) for EATSS increases from 2 to 3, there is a performance

Table 1: Performances by setting different values of $\theta$ and M in HCRAC.

| $\theta$ | M | AP | $AP_{50}$ | $AP_{75}$ |
|---|---|---|---|---|
| 4 | 20 | 37.5 | 56.8 | 40.2 |
| 5 | 20 | **37.6** | 57.2 | 40.1 |
| - | 20 | 37.4 | 57.1 | 39.8 |
| 6 | 20 | 37.4 | 57 | 39.8 |
| 5 | 25 | 37.3 | 56.6 | 39.9 |
| 5 | 30 | 37.5 | 57 | 40.1 |

Table 2: Performances of different values of $\gamma$ in HCRAC.

| $\mu$ | AP | $AP_{50}$ | $AP_{75}$ |
|---|---|---|---|
| 0.6 | 37.3 | 56.8 | 39.7 |
| 0.7 | **37.6** | 57.2 | 40.1 |
| 0.8 | 37.5 | 57.1 | 39.7 |
| 0.9 | 37.5 | 56.8 | 40.2 |

Table 3: Performances of different values of $\alpha$ and ep in HCRAR.

| $\alpha$ | ep | AP | $AP_{50}$ | $AP_{75}$ |
|---|---|---|---|---|
| 0 | 0.01 | 37.2 | 55.7 | 39.2 |
| -0.1 | 0.01 | 37.3 | 55.9 | 39.7 |
| -0.1 | 0.1 | 37.2 | 55.8 | 39.4 |
| -0.1 | 0.001 | **37.4** | 56 | 39.8 |
| -0.2 | 0.01 | 37.2 | 55.7 | 39.7 |
| - | - | 37.1 | 55.6 | 39.6 |

Table 4: Performances of Retinanet of different values of $\gamma$ in HCRAR.

| $\gamma$ | AP | $AP_{50}$ | $AP_{75}$ |
|---|---|---|---|
| 0.8 | 37.1 | 55.8 | 39.7 |
| 1.0 | 37.1 | 55.7 | 39.5 |
| 1.2 | **37.4** | 56 | 39.8 |
| 1.4 | 37.2 | 55.9 | 39.5 |

Table 5: The individual impact of each element based on FCOS+ATSS. (With auxiliary modules)

| EATSS | HCRAR | HCRAC | AP | $AP_{50}$ | $AP_{75}$ |
|---|---|---|---|---|---|
| | | | 41.2 | 58.2 | 44.6 |
| ✓ | | | 41.3 | 58.7 | 44.7 |
| ✓ | ✓ | | 41.7 | 59.3 | 45.4 |
| ✓ | ✓ | ✓ | **42.1** | 59.1 | 45.8 |

Table 6: Performance of different parameter l of EATSS based on FCOS+ATSS.

| l | AP | $AP_{50}$ | $AP_{75}$ |
|---|---|---|---|
| 2 | 41.7 | 58.7 | 45.4 |
| 3 | **42.1** | 59.1 | 45.8 |
| 4 | 41.9 | 59.2 | 45.7 |

improvement of 0.4 AP in Table 6. This confirms that HCRAL has a significant optimization effect on samples with high scores and low IoU or low scores and high IoU, and EATSS's strategy aligns well with HCRAL. Secondly, classification loss is not in terms of focal loss but in terms of HCRAC loss, the performance is improved to 41.7 AP. Finally, GIoU loss is replaced with HCRAR loss, the performance boost to 42.1 AP. These results verify the effectiveness of the method we proposed.

## 4.3 COMPARISON WITH STATE-OF-THE-ART

In Table 7, we compared HCRAL with state-of-the-art loss functions on COCO test-dev. Notably, all loss function use one-stage model with same multi-scale training (Li et al. (2020); Zhang et al. (2021; 2020)). HCRA loss achieves 44.4 AP on FCOS+ATSS with backbone as ResNet-50, exceed all other competitive methods with the same backbone, such as GFL(43.1 AP) and VFL(43.6 AP). We also apply HCRAL into deeper network like ResNet-101, surpassing VFL and GFL by 1.2 AP and 1.1 AP, which show HCRAL verify its great performance. With auxiliary modules, our proposed method achieve higher accuracy than other recent state-of-the-art methods. Furthermore, we use Res2Net (Gao et al. (2021)) and deformable convolution layers (Dai et al. (2017); Zhu et al. (2019b)), resulted in a remarkable performance, achieving a notable AP of 51.4.

## 4.4 GENERALIZATION AND SUPERIORITY OF HCRA

We assessed HCRA loss functions by substituting HCRAC for existing classification loss functions and HCRAR for regression loss functions in popular detectors, RetinaNet and ATSS, using the COCO val2017 dataset. In Table 8, HCRAC achieves a 1.1 AP improvement for RetinaNet (37.6 AP vs. 36.5 AP) and a 0.4 AP improvement for ATSS (40.4 AP vs. 40 AP), which show its

Table 7: Performance (one-stage model) comparison with state-of-the-art detectors on MS COCO test-dev. R : ResNet. R2 : Res2Net. DCN : Deformable convolution network. AUX means adding auxiliary modules.

| Method | Backbone | AUX | Epoch | AP | $AP_{50}$ | $AP_{75}$ | $AP_S$ | $AP_M$ | $AP_L$ |
|---|---|---|---|---|---|---|---|---|---|
| RetinaNet (Lin et al. (2017)) | R-101 | × | 18 | 39.1 | 59.1 | 42.3 | 21.8 | 42.7 | 50.2 |
| FreeAnchor (Zhang et al. (2019)) | R-101 | × | 24 | 43.1 | 62.2 | 46.4 | 24.5 | 46.1 | 54.8 |
| FSAF (Zhu et al. (2019a)) | R-101 | × | 18 | 40.9 | 61.5 | 44.0 | 24.0 | 44.2 | 51.3 |
| FCOS (Tian et al. (2019)) | R-101 | × | 24 | 41.5 | 60.7 | 45.0 | 24.4 | 44.8 | 51.6 |
| SAPD (Zhu et al. (2020)) | R-101 | × | 24 | 43.6 | 62.1 | 47.4 | 26.1 | 47.0 | 53.6 |
| SAPD (Zhu et al. (2020)) | R-101-DCN | × | 24 | 46.0 | 65.9 | 49.6 | 26.3 | 49.2 | 59.6 |
| ATSS (Zhang et al. (2020)) | R-101 | × | 24 | 43.6 | 62.1 | 47.4 | 26.1 | 47.0 | 53.6 |
| ATSS (Zhang et al. (2020)) | R-101-DCN | × | 24 | 46.3 | 64.7 | 50.4 | 27.7 | 49.8 | 58.4 |
| GFL (Li et al. (2020)) | R-50 | × | 24 | 43.1 | 62.0 | 46.8 | 26.0 | 46.7 | 52.3 |
| GFL (Li et al. (2020)) | R-101 | × | 24 | 45.0 | 63.7 | 48.9 | 27.2 | 48.8 | 54.5 |
| GFLv2 (Li et al. (2021)) | R-101-DCN | ✓ | 24 | 48.3 | 66.5 | 52.8 | 28.8 | 51.9 | 60.7 |
| VFL (Zhang et al. (2021)) | R-50 | × | 24 | 43.6 | 62.2 | 47.4 | 26.3 | 47 | 53.5 |
| VFL(Zhang et al. (2021)) | R-101 | × | 24 | 44.9 | 64.1 | 48.9 | 27.1 | 48.7 | 55.1 |
| VFL (Zhang et al. (2021)) | R-101-DCN | ✓ | 24 | 49.2 | 67.5 | 53.7 | 29.7 | 52.6 | 62.4 |
| HCRAL (ours) | R-50 | × | 24 | 44.4 | 62.2 | 48.6 | 26.9 | 47.6 | 55.5 |
| HCRAL (ours) | R-101 | × | 24 | 46.1 | 64.1 | 50.4 | 28.1 | 49.7 | 57.5 |
| HCRAL (ours) | R-101-DCN | ✓ | 24 | 49.3 | 67.3 | 53.9 | 30.4 | 52.8 | 63.1 |
| HCRAL (ours) | R2-101-DCN | ✓ | 24 | **51.4** | **69.5** | **56.1** | **32.5** | **54.9** | **64.9** |

superior performance over other classification loss functions. For regression loss functions, it could be concluded in Table 9 that HCRAR achieved a notable mAP of 37.4 on RetinaNet, outperforming other IoU loss functions, resulting in a significant improvement of 0.9 AP compared to the baseline. While the improvement on ATSS was relatively smaller due to its combination with a rescaling regression weight, it still surpassed Focal EIoU by 0.6 AP.

Table 8: Performance comparison using popular classification losses versus our HCRAC loss on Retinanet and ATSS. † means denotes the default loss function for the baseline.

| Detector | Loss | AP |
|---|---|---|
| | FL† (Lin et al. (2017)) | 36.5 |
| | QFL (Li et al. (2020)) | 37.3 |
| Retina | DR (Qian et al. (2020)) | 37.4 |
| | VFL (Zhang et al. (2021)) | 37.4 |
| | HCRAC (ours) | **37.6** |
| | FL† (Lin et al. (2017)) | 40.0 |
| | QFL (Li et al. (2020)) | 40.2 |
| ATSS | VFL (Zhang et al. (2021)) | 39.8 |
| | HCRAC (ours) | **40.4** |

Table 9: Comparison of performances when applying popular IoU Loss and our HCRAR loss to Retinanet and ATSS. † means denotes the default loss function for the baseline.

| Detector | Loss | AP |
|---|---|---|
| | L1† | 36.5 |
| | EIoU (Zhang et al. (2022)) | 37.0 |
| Retina | F-EIoU (Zhang et al. (2022)) | 37.2 |
| | $\alpha$-IoU (He et al. (2021)) | 36.4 |
| | HCRAR (ours) | **37.4** |
| | GIoU† (Rezatofighi et al. (2019)) | 40.0 |
| | F-EIoU (Zhang et al. (2022)) | 39.6 |
| ATSS | $\alpha$-IoU (He et al. (2021)) | 39.1 |
| | HCRAR (ours) | **40.2** |

## 5 CONCLUSION

In this paper, we propose HCRA loss, a hybrid classification and regression loss function, along with a novel strategy called EATSS for object detection. We incorporate the RCI module, designed to address inconsistencies between classification and regression tasks, and the CF module, which focuses on difficult-to-train samples within each task, enabling the model to concentrate on the most informative training samples. To further assess the effectiveness of our loss function, we incorporate EATSS into dense object detectors and evaluate the performance of our proposed approach. More-

over, we conduct comparative experiments on one-stage detectors, demonstrating the efficacy and generalization of HCRAL.

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

## A   APPENDIX

You may include other additional sections here.

