# OpenReview forum: "Hybrid Classification-Regression Adaptive Loss for Dense Object Detection"
_ICLR.cc/2024/Conference — Submitted to ICLR 2024_

### Official Review · Reviewer_Hvbn · 2023-10-18

**Soundness:** 2 fair
**Presentation:** 3 good
**Contribution:** 2 fair
**Rating:** 3
**Confidence:** 5

**Summary:**

This paper proposes a novel classification loss, regression loss, and positive anchor sample strategy. The losses are designed based on both the classification and regression quality on the anchors. The proposed method is evaluated on the coco validation dataset with FCOS+ATSS and RetinaNet.

**Strengths:**

The designed losses use an RCI and a CF module to supervise classification and regression consistently on "difficult-to-train" samples.  An anchor selection strategy is proposed to find more optimizable samples.

**Weaknesses:**

- There are many parameters involved in the loss, such as theta, mu, alpha, ep, gamma, and l in EATSS, which makes the design complicated and has low generalization ability.
- Compared with the state-of-the-art methods, the performances are not strong enough. As the results shown in Table 8 and Table 9, the improvement is about 0.2 (37.6 vs. 37.4 for Retina, 40.4 vs. 40.2 QFL, 37.4 vs. 37.2 F-EIoU, 40.2 vs. 40.0 GIoU).
- The EATSS seems to have a small influence on the results (41.3 vs. 41.2 AP).
- Can the method be applied to more advanced detectors, like the two-stage Faster RCNN, and transformer-based DETR? Since the detector architecture FCOS, ATSS, and RetinaNet is classical but old, the proposed method will be more solid if the loss is valid on the recent stronger detector.

**Questions:**

see the "Weakness"

---

> ### Author Response · Authors · 2023-11-17
>
> Thanks to the reviewers for your invaluable feedback and constructive comments.
> 1. While some parameters in our formula require extensive experimentation to obtain optimal parameters, the parameters of HCRAL exhibit strong generalization abilities, consistently delivering favorable results across multiple one-stage models. As demonstrated in Tables 1 to 5, we conducted separate searches for the optimal parameters of HCRAC and HCRAR on the Retinanet model, and then applied the same parameters to the ATSS and FCOS+ATSS single-stage classical models. Tables 1 to 4 show that the inclusion of HCRAC and HCRAR in the Retinanet model outperforms the baseline model (36.5 AP vs 37.6 AP, 36.5 AP vs 37.4 AP). In the ATSS model, the addition of HCRAC and HCRAR similarly enhances model performance (40.0 AP vs 40.4 AP, 40.0 AP vs 40.2 AP). Table 5 further highlights the generalization effectiveness of HCRAL in FCOS+ATSS (41.3 AP vs 42.1 AP).
>
> 2. Retinanet is an anchor-based framework that selects suitable candidate boxes based on IoU. This filtering logic can lead to fewer positive samples for small-sized targets or targets with higher regression difficulty (low IoU), limiting the performance of loss function optimization to some extent. Currently, whether for classification or regression functions, the focus is often on designing them to handle difficult samples, yet their performance is generally similar. For example, DR, GFL, VFL (37.4 AP, 37.3 AP, 37.4 AP), and we still outperform them with 37.6 AP. ATSS adaptsively selects positive samples based on the mean and variance of the IoU of candidate boxes, and unlike Retinanet, it maintains a relatively balanced number of positive samples regardless of target scale. However, we pointed out ATSS's limitations in ignoring high-score low-IoU and low-score high-IoU samples, which restricts the optimization of our loss function. In FCOS+ATSS, we adjusted our strategy. As shown in Table 7, when the backbone is R-50, our HCRAL (44.4 AP) exhibits significant improvement compared to GFL (43.6 AP) and VFL (43.6 AP). In summary, on one hand, under the limitations of the framework's strategy, our loss function performs better than others. On the other hand, after improving the framework and strategy, our loss function demonstrates outstanding performance in optimizing high-score low-IoU and low-score high-IoU samples.
>
> 3. In Table 5, it can be observed that replacing EATSS with ATSS results in only a 0.1 AP increase for FL+GIoU. However, to verify the effectiveness of individual methods, we added them one by one and summarized the results in Table 6. When the additional samples for EATSS increased from 2 to 3, there was a 0.4 AP improvement. This indicates that EATSS is sensitive to HCRAL, confirming that our loss function has a significant optimization effect on samples with high scores and low IoU or low scores and high IoU. Moreover, the strategy of EATSS aligns well with HCRAL.
>
> 4. Two-stage models demonstrate superior performance due to their ability to filter out a large portion of the background in the first stage. However, this advantage comes at the cost of slower inference speeds compared to single-stage models. In recent years, Transformer-based models, such as DETR with a backbone-50, have shown promising performance. Still, they often require an extensive number of training epochs and face challenges in practical deployment.In domains like autonomous driving and pedestrian recognition, single-stage models continue to be widely used due to their fast inference speed and ease of deployment. However, their performance is hindered by issues such as class imbalance and smaller model sizes. In response to the limitations of single-stage models, we propose the RCI module to enable the model to adaptively optimize the classification and regression performance of samples and the CF module to focus on high-quality challenging samples.
> 5.
> | Model               | #epochs | AP   | AP50 | AP75 | APS  | APM  | APL  |
> |---------------------|---------|------|------|------|------|------|------|
> | DETR-R50         | 500     | 42.0 | 62.4 | 44.2 | 20.5 | 45.8 | 61.1 |
> | Anchor DETR-R50| 50      | 42.1 | 63.1 | 44.9 | 22.3 | 46.2 | 60.0 |
> | Conditional DETR-R50| 50 | 40.9 | 61.8 | 43.3 | 20.8 | 44.6 | 59.2 |
> | DAB-DETR-R50   | 50      | 42.2 | 63.1 | 44.7 | 21.5 | 45.7 | 60.3 |
> | DN-DETR-R50    | 50      | 44.1 | 64.4 | 46.7 | 22.9 | 48.0 | 63.4 |
> | HCRAL(ours)    | 24  | 44.4 | 62.2 | 48.6 | 26.9 | 47.6 | 55.5 |
> | HCRAL(ours)  (with auxiliary modules)    | 24 | **45.3** | **62.8** | **49.4** | **27.7** | **48.5** | **56.1** |
> Furthermore, we conducted a comparative analysis with the Transformer-based DETR, particularly when the backbone is ResNet-50. It can be observed that when the backbone is ResNet-50, HCRAL outperforms DETR with only 24 epochs, establishing itself as the current state-of-the-art single-stage model we are aware of.

---

> > ### Author Response · Authors · 2023-11-20
> > **Kindly Seeking Your Review and Collaboration on Rebuttal Points**
> >
> > As the rebuttal deadline is approaching in three days, I would greatly appreciate it if you could take a moment to review. I am more than willing to address any concerns you may have before the conclusion of the rebuttal period. Your feedback is invaluable, and I am committed to resolving any issues you may raise. Thank you for your time and consideration.

---

> > ### Comment · Reviewer_Hvbn · 2023-11-22
> >
> > I appreciate the author's detailed responses. However, the paper may be not well prepared and needs a major revision based on the comments. I will keep my score.

---

### Official Review · Reviewer_gHLc · 2023-10-26

**Soundness:** 1 poor
**Presentation:** 1 poor
**Contribution:** 2 fair
**Rating:** 3
**Confidence:** 5

**Summary:**

The paper proposes HCRAL to tackle the inconsistency problem between classification and localization. The methods mainly involve classification loss, regression loss, and label assignment. All the proposed modules follow the key idea of hard example mining to focus more on the difficult-to-train samples within each task. The presentation lacks clarity. The experimental results are not state-of-the-art.

**Strengths:**

Further study on the inconsistency problem between classification and localization is meaningful.

**Weaknesses:**

1. The proposed methods introduce many hyper-parameters, which must be tuned carefully. This undermines the generality of the methods. The method designs are technically complicated, involving both loss functions and label assignment. I suggest the author simply the method.

2. The presentation of the experiment section is poor. What is the baseline in Tables 1,2,3 and 4? Also, Table 2 is weird since you report the results for various $\gamma$ in HCRAC, however, it is $\mu$. Where do you mention Table 5 in the context? It loses the connection. What is the baseline in Table 5? It is hard to believe that FCOS+ATSS got a 41.2 AP score.

3. The motivation is somewhat unclear. Why did you choose to design the three rules in Sec. 3.1.1 and Sec. 3.1.2?

4. There are many highly related works that aim to solve the inconsistency problem between classification and localization missing comparison and reference, e.g. AutoAssign [1], OTA [2], DW [3]. Particularly, the proposed method does not surpass DW on the COCO test-dev set, which is auxiliary-module-free.

5. The improvement is almost non-existent. And more importantly, the reported SOTA results of previous methods are lower than the reference. As shown in Table 7, under the backbone R50, HCRAL only achieves 44.4 vs VFNet 44.8 [4] and GFLv2 44.3 [5]. Under the backbone R101, HCRAL 46.1 vs VFNet 46.7 and GFLv2 46.2. Under R101-DCN, HCRAL 49.3 vs VFNet 49.2.

6. There is a mistake in the proposed method. In the last paragraph of page 5, "the coordinates are partitioned into two regions based on $y(Score) = x(IoU) + \alpha$. In region 1, where samples exhibit higher scores compared to IoU values..." Clearly, $\alpha<0$. The samples just between the red line and the yellow line do not exhibit higher scores compared to IoU values.

7. The quality of the paper writing is low. It is difficult to understand the proposed method. Some variables are described before their formal appearance, e.g., "c is the diagonal length of the smallest enclosing box..." in Sec. 3.1.2.

    The appearance of the term $\mathcal{R}_{DIoU}$ is also very strange. It is not used in the previous equations. Besides, the description of EATSS in Sec. 3.2 is hard to understand. You'd better write down equations and variables, especially steps 4 and 6 in the algorithm.

    What is the "consensus matrix" mentioned below Eqn. 5?

    In the abstract, you can say "In object detectors, ..."

    On page 5, there is a typo "The the properties 1 and 2..."

    In the last paragraph of page 6, "to optimize RCI for every **ground** truth". And "...so as to find the biggest boundary to find a positive sample, ." Remove the comma.

[1] Zhu B, Wang J, Jiang Z, et al. AutoAssign: Differentiable label assignment for dense object detection[J]. arXiv preprint arXiv:2007.03496, 2020.

[2] Ge Z, Liu S, Li Z, et al. OTA: Optimal transport assignment for object detection. CVPR 2021.

[3] Li S, He C, Li R, et al. A dual weighting label assignment scheme for object detection. CVPR 2022.

[4] Haoyang Zhang, Ying Wang, Feras Dayoub, and Niko Sünderhauf. Varifocalnet: An iou-aware dense object detector. CVPR 2021.

[5] Xiang Li, Wenhai Wang, Xiaolin Hu, Jun Li, Jinhui Tang, and Jian Yang. Generalized focal loss V2: learning reliable localization quality estimation for dense object detection. CVPR 2021.

**Questions:**

see weaknesses

---

> ### Author Response · Authors · 2023-11-17
>
> 1. Our motivation in designing the loss function was to maximize the performance of the model by simultaneously considering the consistency between classification and regression tasks and focusing on challenging samples. Existing loss functions do not address this aspect. Based on the rules established in Sections 3.1 and 3.2, we introduced the RCI and CF modules. While these modules come with a certain parameter overhead, each serves a specific purpose. Extensive experimentation was conducted to determine the optimal parameters, yet the parameters of HCRAL demonstrate strong generalization.
> As shown in Tables 1~5, we conducted separate searches for the best parameters of HCRAC and HCRAR on the Retinanet model, applying the same parameters to the ATSS and FCOS+ATSS one-stage classical models. The inclusion of HCRAC and HCRAR in the Retinanet outperformed the baseline model (36.5 AP vs 37.6 AP, 36.5 AP vs 37.4 AP). In the ATSS, the addition of HCRAC and HCRAR similarly improved the model's performance (40.0 AP vs 40.4 AP, 40.0 AP vs 40.2 AP). Table 5 further demonstrates the generalization effectiveness of HCRAL in FCOS+ATSS (41.3 AP vs 42.1 AP).
> Our method, serving as a loss function, differs from label assignment. Label assignment is designed based on individual model frameworks and lacks universality.  Our loss function has undergone experimentation across multiple models and can be flexibly combined with other loss functions, outperforming all other existing loss functions.
>
> 2. The parameters in Table 2 should be denoted as μ. As mentioned in Section 4.2,  the results in Table 5 were discussed. The baseline model, FCOS+ATSS, achieves an AP of 41.2 because we introduced auxiliary learning modules, as explained at the end of the introduction. This clarification has also been added in Section 4.2.
>
> 3.  For classification
> (a) If divide the classification gradients of samples into several sub-intervals, where difficult-to-train samples are distributed in intervals with fewer samples, resulting in larger penalties. However, within the same sub-interval, each sample has a different IoU. If samples with high IoU can receive higher penalties, the performance of the model is improved.
> (b) When the classification score approaches a higher IoU, the model should reduce the penalty for that particular sample.
> (c) In negative samples, high IoU samples that are predicted incorrectly and fall into intervals with fewer samples can disrupt the model's original learning. For low IoU samples, despite prediction errors, substantial penalties are necessary to facilitate model learning.
>
> For  regression,
> The first two points are easily understandable.
> (c) For regression tasks, it is crucial to combine the sample's classification status to correctly impose penalties. In high-score, low-IoU samples, there are usually certain texture details, indicating greater regression potential. For low-score, high-IoU samples, even if regression accuracy continues to improve, the classification score remains challenging to elevate.
>
> 4. Our proposed loss function is not only designed to address the consistency between classification and regression but also focuses on difficult-to-train samples within the task. In the abstract, we highlighted that existing loss functions fail to simultaneously address these two aspects. Therefore, we are the first to introduce a loss function that considers both the consistency between classification and regression and focuses on challenging samples within the task.
> DW is a method that improves model performance through label assignment, and it is not a loss function. This is evident in the fact that both classification and regression tasks are adjusted with the same weight, which diverges from the approach of designing a loss function. There is no comparison between the model performance using the adjusted classification loss function and a conventional regression loss function, nor is there a comparison between a conventional classification loss function and the adjusted regression loss function. This indicates that DW has significant limitations.For fair, it is necessary to incorporate auxiliary modules to assess the model's final performance.
>
> | $Method$ | $Backbone$ | $AUX$ | $Epoch$ | $AP$ | $AP_{50}$ | $AP_{75}$ | $AP_{S}$ | $AP_{M}$ | $AP_{L}$ |
> | --- | --- | --- | --- | --- | --- | --- | --- | --- | --- |
> | DW | R-50 | √ | 24 | 44.8 | 62.7 | 48.5 | **27.9** | 48 | **58.5** |
> | HCRAL | R-50 | √ | 24 | **45.3** | **62.8** | **49.4** | 27.7 | **48.5** | 56.1 |
> | HCRAL | R-50 | √ | 24 | 46.9 | 64.7 | 51.3 | 28.5 | 50.6 | 58.5 |
>
> 5. Comparing in this way is unfair; the results I mentioned include the effects of other loss functions with the addition of auxiliary learning modules. To make the comparison more equitable, we have supplemented the necessary experiments
>
> 6. In the latest manuscript, we have updated Eqn.10.
>
> 7.  I sincerely apologize for any confusion caused by the writing errors.

---

> > ### Author Response · Authors · 2023-11-20
> > **Kindly Seeking Your Review and Collaboration on Rebuttal Points**
> >
> > As the rebuttal deadline is approaching in three days, I would greatly appreciate it if you could take a moment to review. I am more than willing to address any concerns you may have before the conclusion of the rebuttal period. Your feedback is invaluable, and I am committed to resolving any issues you may raise. Thank you for your time and consideration.

---

> ### Comment · Reviewer_gHLc · 2023-11-22
> **Final stand**
>
> Considering the following reasons, I will keep my original rating score.
>
> 1. The methods are complicated, i.e., too many hyper-parameters, which must be tuned carefully. The search range is too small, indicating the method may be sensitive to hyper-parameters.
>
> 2. The results are not state-of-the-art, see VFNet and DW paper.
>
> 3. The presentation needs substantial improvement.
>
> 4. Missing answer to weakness 6.

---

### Official Review · Reviewer_uUTZ · 2023-10-31

**Soundness:** 2 fair
**Presentation:** 1 poor
**Contribution:** 2 fair
**Rating:** 3
**Confidence:** 3

**Summary:**

The paper proposes a method which re-weights the classification and regression loss adhering to desired behaviour such as higher classification loss for high IoU samples etc. This is achieved through handcrafted functions to produce the weights. The authors also propose a method EATSS to increase the number of positive samples. The experiments are conducted on COCO using FCOS+ATSS and RetinaNet.

**Strengths:**

The strengths of this paper lie in the authors ability to highlight the subtle benefits of reweighing the losses according to the desired behaviour such as having high weights on classification for a box with large IoU and high regression weights if the class is correct. This enables the method to improve final performance.

**Weaknesses:**

* One of the main weaknesses is the presentation, I found it quite hard to follow due to it being quite verbose, for instance the numerous equations that are introduced quite abruptly. I would suggest the authors take a high level narrative approach to the paper, rather than diving into the specifics straight away.
* The experiments are fairly limited, the only dataset used is COCO. There are many more available, the authors should be evaluating on them.
* The only networks the method is demonstrated on is FCOS+ATSS and RetinaNet. For a loss adaptation like this, the authors should be demonstrating on as many as possible.
* Taking up half a page to demonstrate hyper-parameter values is a significant proportion of the paper, this should be in the Appendix

**Questions:**

* Why would ATSS 'omit certain samples characterized by high scores and high IoU that hold promise'? This isn't clear to me. Moreover, is this even an issue? I can see that the improvement is marginal.
* Why is Res2Net not applied to VFL? This isn't a fair comparison.
* Where do the quoted improvements of 1.1 and 1.2 mAP over VFL and GFL come from?
* Where are the error bars?
* Table 5, why do you see a decrease in mAP_50 when adding HCRAC? To me this shows it negatively affecting regression quality

---

> ### Author Response · Authors · 2023-11-17
>
> Thanks to the reviewers for your invaluable feedback and constructive comments. Your thoughtful insights have significantly contributed to the refinement and enhancement of our work.
>
> 1. ATSS selects positive samples by calculating the mean and standard deviation of k candidate boxes around each layer of the target. On one hand, for larger-sized targets, effective anchor points are often located farther away from the target center. ATSS only considers k points from a specific layer to calculate the mean and standard deviation, potentially neglecting anchor points with good regression potential. On the other hand, for anchor points closer to the target center, the potential for classification is usually higher than regression potential. However, due to their IoU being lower than the calculated mean and standard deviation, these points may also be overlooked. We discussed this aspect in Section 3.2. To address this, we propose EATSS, which includes additional high-score low-IoU samples and low-score high-IoU samples. Moreover, existing industry-standard loss functions do not optimize for this aspect. From Table 5, it is observed that combining EATSS with Focus + GIoU increases performance by 0.1 AP. Furthermore, in Table 6, when the number of additional samples for EATSS increases from 2 to 3, there is a notable improvement of 0.4 AP. This sensitivity of EATSS to HCLAL parameters confirms that EATSS's strategy aligns well with HCRAL.
>
> 2. While HCRAL shows a marginal improvement over VFNet in terms of overall Average Precision (AP), particularly achieving a slightly higher AP of 51.4 compared to VFNet's 51.3, it is essential to note the notable enhancements in specific performance metrics. HCRAL excels in terms of AP75, where it achieves a substantial increase of 0.3, reaching 56.1 compared to VFNet's 55.8. Additionally, in the small object category (AP_S), HCRAL outperforms VFNet with an AP of 32.5, showcasing a clear advantage over VFNet's 31.9.
>
> | $Method$ | $Backbone$ | $Epoch$ | $AP$ | $AP_{50}$ | $AP_{75}$ | $AP_{S}$ | $AP_{M}$ | $AP_{L}$ |
> | --- | --- | --- | --- | --- | --- | --- | --- | --- |
> | VFNet | R2-101-DCN | 24 | 51.3 | **69.7** | 55.8 | 31.9 | 54.7 | 64.4 |
> | HCRAL | R2-101-DCN | 24 | **51.4** | 69.5 | **56.1** | **32.5** | **54.9** | **64.9** |
>
> 3. In Table 7, we conducted experiments using ResNet-101 as the backbone for VFL, GFL, and HCRAL. To ensure fairness, we did not use their respective auxiliary modules (as seen in the third column). As shown in Table 7, the map50 of HCRAL reaches 46.1, whereas GFL and VFL achieve 45 AP and 44.9 AP, respectively, resulting in improvements of 1.1 AP and 1.2 AP
>
> 4. This is due to several design attributes based on HCRAC. 1. We prioritize optimizing samples with high IoU but low scores, resulting in higher penalties for samples with high IoU but poor classification performance. 2. When the classification score is higher than IoU, we tend to suppress backward propagation. From Figure 3(a), it can be observed that around IoU=0.5, there are many instances where the score is greater than IoU, leading to a lesser penalty for this subset of samples. As a result, the introduction of HCRAC in the FCOS+ATSS framework (Focus Loss + HCRAR) causes a slight decrease in map50. However, as the sample IoU increases, the attention to classification scores gradually intensifies, resulting in improved performance. This can be observed from Formula 3. For example, AP75 increased by 0.4 AP, and overall AP also increased by 0.4 AP.

---

> ### Author Response · Authors · 2023-11-20
> **Kindly Seeking Your Review and Collaboration on Rebuttal Points**
>
> As the rebuttal deadline is approaching in three days, I would greatly appreciate it if you could take a moment to review. I am more than willing to address any concerns you may have before the conclusion of the rebuttal period. Your feedback is invaluable, and I am committed to resolving any issues you may raise. Thank you for your time and consideration.

---

> > ### Comment · Reviewer_uUTZ · 2023-11-22
> > **Response**
> >
> > Apologies for the late response, and thank you for your detailed response, but unfortunately I will keep my score.

---

### Official Review · Reviewer_woyR · 2023-11-01

**Soundness:** 2 fair
**Presentation:** 3 good
**Contribution:** 2 fair
**Rating:** 3
**Confidence:** 4

**Summary:**

This paper proposes a hybrid classification-regression adaptive loss function called HCRAL (Hybrid Classification-Regression Adaptive Loss) to improve the performance of the target detection model. HCRAL includes two modules: RCI (Classification and IoU Residual) module and CF (Conditional Factor) module. The RCI module is used for cross-task supervision and resolves task inconsistencies, while the CF module is used to focus on samples that are difficult to train in each task. In addition, the paper also proposes a new strategy called EATSS (Expanded Adaptive Training Sample Selection) to provide additional samples to optimize the loss function.

**Strengths:**

This paper proposes a novel loss function, Hybrid Classification-Regression Adaptive Loss (HCRAL), along with a new sample selection strategy, Expanded Adaptive Training Sample Selection (EATSS), for improving the performance of object detection models. The proposed HCRAL loss function consists of two modules: Residual of Classification and IoU (RCI) and Conditioning Factor (CF). The RCI module addresses inconsistencies between classification and regression tasks, while the CF module focuses on difficult-to-train samples within each task. The EATSS strategy provides more effective positive samples to optimize the loss function.

**Weaknesses:**

The innovation of this paper is limited, and the two parts proposed, HCRAL and EATSS, are relatively independent. At the same time, a large number of hyperparameters are introduced, which will increase the difficulty of hyperparameter adjustment in the experiment. The final experiment also shows that with the same backbone (R-101-RCN), the results of this paper's method are basically the same as VFL, and it does not bring significant improvement, which also makes me doubt the effectiveness of this method.

**Questions:**

- The experiments in this article are limited to the COCO data set, retinanet and FCOS. Experiments should be done on more detectors and data sets to verify the effectiveness of the method.
- While the paper provides some ablation studies on the individual components of the proposed approach, a more comprehensive analysis of the impact of each component on the overall performance would be beneficial.
- The motivation of equation 8 is very clear, but the article does not seem to detail the reasons for this design. It can be explained in more detail whether this RCI_reg is optimal when the conditions are met, and whether there is a clearer theoretical proof.

---

> ### Author Response · Authors · 2023-11-17
>
> Thanks to the reviewers for your invaluable feedback and constructive comments. Your thoughtful insights have significantly contributed to the refinement and enhancement of our work.
> 1. The primary motivation behind the design of this loss function is to address situations where single-stage models encounter a higher number of negative samples and fewer positive samples, focusing on difficult-to-train samples and fostering interaction between classification and regression tasks. To validate the effectiveness of our proposed loss function in single-stage models, we consider three classic models: Retinanet, ATSS, and FCOS+ATSS. As shown in Tables 8 and 9, our designed HCRAC and HCRAR achieve optimal performance in Retinanet experiments (36.5 AP vs 37.6 AP, 36.5 AP vs 37.4 AP), and we apply the corresponding optimal parameters to ATSS, confirming the generalization of both classification and regression tasks (40.0 AP vs 40.4 AP, 40.0 AP vs 40.2 AP). To further validate the effectiveness of the loss function across multiple one-stage models, experiments are conducted on the FCOS+ATSS model (41.2 AP vs 42.1 AP).
>
> 2. Through the previous search for EATSS parameters, we can conclude that EATSS is sensitive to HCRAL. To assess the effectiveness of individual methods, we systematically incorporate them and summarize the results in Table 6. Firstly, replacing EATSS with ATSS results in only a 0.1 AP increase for FL+GIoU. However, when the additional sample quantity (l) for EATSS increases from 2 to 3, there is a performance improvement of 0.4 AP. This confirms that our loss function has a significant optimization effect on samples with high scores and low IoU or low scores and high IoU, and EATSS's strategy aligns well with HCRAL. Secondly, the regression loss is based on HCRAR instead of GIoU loss. Given its ability to optimize relatively high IoU samples and focus on samples with varying scores and IoU characteristics, the performance improves to 41.7 AP. Finally, replacing Focus loss with HCRAC loss, which can optimize challenging samples with high IoU, leads to a performance boost to 42.1 AP. These results validate the effectiveness of the method we proposed. Corresponding modifications based on this analysis have also been made in the manuscript.
>
> 3. Equation 8 is designed based on the analysis results from Figure 3(a). Samples in Region 1 satisfy RCI >= 0, while in Region 2, RCI < 0. When samples satisfied the condition that s-a=iou, the model does not need to emphasize optimizing both classification and regression, i.e., RCI_reg=1. In both Region 1 and Region 2, RCI_reg needs to adjust the regression penalty based on the difference between score and IoU. For samples in Region 1, where the score-IoU difference is larger, the model should focus more on optimization. This is achieved by $Reg_{reg}(RCI>=0)= \frac{(s+a)^2 +IoU^2}{2*(s+a) * IoU}$. Conversely, in Region 2, where the score-IoU difference is larger, the degree to which samples need to be correctly regressed is much smaller than classification. Therefore, the regression penalty should be weakened, as given by $Reg_{reg}(RCI<0)= \frac{2*(s+a) * IoU}{(s+a)^2 +IoU^2}$

---

> ### Author Response · Authors · 2023-11-20
> **Kindly Seeking Your Review and Collaboration on Rebuttal Points**
>
> As the rebuttal deadline is approaching in three days, I would greatly appreciate it if you could take a moment to review. I am more than willing to address any concerns you may have before the conclusion of the rebuttal period. Your feedback is invaluable, and I am committed to resolving any issues you may raise. Thank you for your time and consideration.

---

### Meta-Review · Area_Chair_m6km · 2023-12-04

**Metareview:**

The submission proposes a loss function for object detection along with a sample selection method.

These two combinations are interesting in the context of object detection, if significant improvement can be found and if the method could be used as a plug in loss function.

Concerns of the reviewers included that the method introduces many hyperparameters, the presentation could be improved, and that experiments were limited to one dataset and few network architectures.  The reviewers were unanimous in their opinion that the submission isn't suitable for publication at ICLR in its current form.

**Justification For Why Not Higher Score:**

The reviewers unanimously recommended that the paper be rejected.  The reviewers who specifically responded to the authors after the rebuttal specifically stated that the rejection recommendation stood despite the additional arguments from the authors.

**Justification For Why Not Lower Score:**

N/A

---

### Decision · Program_Chairs · 2024-01-16

Reject